# Enhancing Graph Routing Algorithm of Industrial Wireless Sensor Networks Using the Covariance-Matrix Adaptation Evolution Strategy

**DOI:** 10.3390/s22197462

**Published:** 2022-10-01

**Authors:** Nouf Alharbi, Lewis Mackenzie, Dimitrios Pezaros

**Affiliations:** 1School of Computing Science, University of Glasgow, Glasgow G12 8LT, UK; 2School of Computing Science, Taibah University, Madinah 42353, Saudi Arabia

**Keywords:** industrial internet of things, industry 4.0, industrial wireless sensor networks, WirelessHART, graph routing, optimisation techniques, covariance-matrix adaptation evolution strategy, best path

## Abstract

The emergence of the Industrial Internet of Things (IIoT) has accelerated the adoption of Industrial Wireless Sensor Networks (IWSNs) for numerous applications. Effective communication in such applications requires reduced end-to-end transmission time, balanced energy consumption and increased communication reliability. Graph routing, the main routing method in IWSNs, has a significant impact on achieving effective communication in terms of satisfying these requirements. Graph routing algorithms involve applying the first-path available approach and using path redundancy to transmit data packets from a source sensor node to the gateway. However, this approach can affect end-to-end transmission time by creating conflicts among transmissions involving a common sensor node and promoting imbalanced energy consumption due to centralised management. The characteristics and requirements of these networks encounter further complications due to the need to find the best path on the basis of the requirements of IWSNs to overcome these challenges rather than using the available first-path. Such a requirement affects the network performance and prolongs the network lifetime. To address this problem, we adopt a Covariance-Matrix Adaptation Evolution Strategy (CMA-ES) to create and select the graph paths. Firstly, this article proposes three best single-objective graph routing paths according to the IWSN requirements that this research focused on. The sensor nodes select best paths based on three objective functions of CMA-ES: the best Path based on Distance (PODis), the best Path based on residual Energy (POEng) and the best Path based on End-to-End transmission time (POE2E). Secondly, to enhance energy consumption balance and achieve a balance among IWSN requirements, we adapt the CMA-ES to select the best path with multiple-objectives, otherwise known as the Best Path of Graph Routing with a CMA-ES (BPGR-ES). A simulation using MATALB with different configurations and parameters is applied to evaluate the enhanced graph routing algorithms. Furthermore, the performance of PODis, POEng, POE2E and BPGR-ES is compared with existing state-of-the-art graph routing algorithms. The simulation results reveal that the BPGR-ES algorithm achieved 87.53% more balanced energy consumption among sensor nodes in the network compared to other algorithms, and the delivery of data packets of BPGR-ES reached 99.86%, indicating more reliable communication.

## 1. Introduction

As one of the key components of the Fourth Industrial Revolution (Industry 4.0) and the Industrial Internet of Things (IIoT) [1], IEEE 802.15.4-based Industrial Wireless Sensor Networks (IWSN) are a promising paradigm for smart industrial automation, due to their advantages of flexibility, low deployment costs and self-organising capabilities. They can potentially significantly improve industrial efficiency and productivity at sites such as oil refineries, steel mills, and chemical plants [2]. WirelessHART, ISA 100.11a, and WIA-PA are the three major industrial communication standards designed for Process Automation (PA) in IWSN applications. As they are intended for industrial automation, they have stringent requirements with regard to communication reliability, balance of energy consumption and end-to-end transmission time [1,2]. A typical IWSN infrastructure based on IEEE 802.15.4 operating in the 2.4 GHz ISM band consists of battery-powered wireless sensor nodes connected through various Access Points (APs) to a gateway as a local destination [2,3]. The Gateway $\left(G_{w}\right)$ establishes a connection with the network control system for the plant automation, which is referred to as the Network Manager (NM). The NM is accountable for network configuration, communication scheduling between network devices, routing management, and system health monitoring and reporting [2]. Centralisation of the IWSN allows better control of network operations and reduces the cost of devices [4].

Routing is an essential task of the NM, and the routes it builds are key to the goals of reliability, latency and balanced energy consumption [2,4]. All IWSN standards specify routing algorithms of two types, Source Routing (SR) and Graph Routing (GR), of which the latter is the more widely used and is exclusively considered herein. GR employs a first-path approach with path redundancy, to transmit data packets from a source node to a $G_{w}$ [2,5]. There are, however, some challenges, as outlined below.

Firstly, industrial environments often generate high levels of noise which may lead to a decline in performance of the routing algorithm [6]. However, the reliability of wireless communication can be improved using multi-channel Time Division Multiple Access (TDMA) and channel hopping. Where sufficient reliability cannot be achieved by the MAC layer [7], redundant routes can be applied by the GR algorithm at the network layer [2]. Retransmission is an effective method for increasing reliability, but it also increases end-to-end transmission time [8].

Secondly, industrial automation imposes stringent end-to-end delay requirements on data communication. Such delays are increased further by conflicts between transmissions where two paths share a sensor node (sender or receiver) [9]. IWSNs do not permit multiple transmissions to take place simultaneously on the same channel; hence, a channel can only support one transmission at a time across the network. A conflict delay occurs when a data packet is delayed because it conflicts with another data packet that is scheduled in the current time slot [9].

Lastly, the workload of sensor nodes around a $G_{w}$ must also be considered since, due to centralisation in IWSNs, nodes closer to the $G_{w}$ are often overburdened with high traffic loads as compared to those further away. This is because packets from the entire region are forwarded through the former to reach the $G_{w}$, leading to an imbalance in energy consumption that reduces the life-time of the network [8].

When designing the best path for a routing algorithm for IWSN, all of these challenges must, hence, be addressed through striking a balance between them. For example, when monitoring systems are used in the industrial domain, sensor nodes with limited power are used in real-world IWSN. This includes monitoring of nuclear plants and furnaces, which could be dangerous applications. Multiple functions may be carried out by sensor nodes. For example, in a temperature monitoring system, the alerting objective is essentially non-critical; however, if the monitored temperature exceeds a certain level, the alerting system may be required to function as a safety system, placing additional demands on the sensor nodes, particularly those near the gateway. As a result, balancing the energy consumption of sensor nodes, increasing communication reliability, and reducing delay are essential requirements of real-life IWSNs. However, these requirements can be relatively difficult to achieve due to interference and noise in industrial environments, which cause constant redundancy, high latency as a result of redundancy, and unbalanced energy consumption.

To address them adequately, optimisation or high-level procedure algorithms are required. The use of optimisation techniques for creating and selecting best paths in a centralised manner may thus be useful for IWSN and future IIoT protocols.

The main optimisation techniques currently used include Swarm Intelligence (SI), Evolution Strategies (ES), and physical-based algorithms [10]. Optimisation techniques typically set an objective function to find the best solution subject to specific criteria [10]. Path optimisation techniques thus play an important role in IWSN, as best routing can promote balanced energy consumption, reduced end-to-end transmission time, and improved network reliability [11].

However, many traditional path optimisation methods are based on Dynamic Programming (DP), which uses a Breadth-First Search (BFS), and Dijkstra which only considers path length to find a best path [5,12,13]. These traditional routing techniques are good for obtaining best solutions, but they each focus on just one requirement of IWSNs and ignore the rest. Dynamic programming is also difficult to link to complex routing problems [10].

Nevertheless, previous research (e.g., [10,11]) has also shown that optimisation techniques are useful for finding the best routing in Wireless Sensor Networks (WSNs), of which IWSNs are a special case [14]. The aim of these path optimisation techniques is to find reliable paths which are energy-efficient [11] by creating an objective function to balance the predetermined requirements which, in the case of IWSNs, are energy consumption, communication reliability, and end-to-end transmission time [14].

The main contribution of this research is the development of a graph routing algorithm of IWSNs based on a Covariance-Matrix Adaptation Evolution Strategy (CMA-ES) [15]. To the best of our knowledge, this the first GR algorithm that specifically adopts evolution strategies to select best paths for IWSNs. This article’s GR algorithm focuses specifically on GR in WirelessHART networks. GR creates paths in a mesh topology, with path redundancy and multi-hop providing additional network reliability in industrial environments. CMA-ES was employed to select the best paths in this form of GR. This is a state-of-the-art optimisation technique in terms of evolutionary computation based on population methods. It has, therefore, been adopted as a standard tool for continuous optimisation in many research laboratories [16] and industrial environments worldwide.

Firstly, the current research proposes three best paths of GR with single-objective functions, depending on the Euclidean distance between sensor nodes (which this article calls PODis), their residual energy (which this article calls POEng) and actual end-to-end transmission time for each data packet between the transmitter and receiver based on the propagation model in the WirelessHART network (called POE2E in this article). The best receiver node for each hop along the best paths of all objective functions is carefully chosen on the basis of a Shortlist, which retains a list of the neighbours of each sensor node within its effective communication range in the $G_{w}$ direction. As a result, this helps to reduce overheads on the network and the energy required to maintain live sensor nodes throughout the entire network.

Secondly, after computing these objective functions, it is necessary to converge on the best solution by means of the proposed algorithm which we call best Path Graph Routing with CMA-ES (BPGR-ES), which uses multiple-objectives to select the final best path. This approach, which constitutes BPGR-ES, can be compared with best single-objective paths (PODis, POEng and POE2E) and existing uplink routing algorithms, using the following performance metrics: average Energy Imbalance Factor (EIF); Packet Delivery Ratio (PDR); Packet Miss Ratio (PMR); total consumed energy and End-to-End Transmission (E2ET).

The remainder of this article is structured as follows: Section 2 presents the background and literature review, which includes an analysis of the state-of-the-art research on graph routing algorithms in IWSNs, optimisation techniques applied to state-of-the-art routing algorithms in wireless networks, and an overview of CMA-ES; Section 3 provides a detailed description of the model of best paths for graph routing based on CMA-ES selection, which is the main focus of this research; Section 4 presents the simulation setup and performance evaluation; Section 5 concludes the article.

## 2. Background and Literature Review

### 2.1. Graph Routing Algorithms

Several algorithms have been proposed for GR to improve adherence to the tight requirements of IWSN. According to [13], energy usage in WirelessHART can be effectively balanced by creating a pre-emptive Energy-Balanced Graph-Routing algorithm (EBGR) for the network node. The suggested algorithm initially applies a BFS set of rules to separate the network into different levels. Subsequently, a graph-routing algorithm re-distributes the energy usage to nodes that have fewer routing activities, by decreasing the links to the nodes that have more such activities. The EBGR approach enhances energy usage and improves network lifetime but can lead to increased overhead across the entire network, since the created graphs are re-structured in each round. This is inefficient compared to other approaches, such as the Enhanced Least-Hop First Routing (ELHFR) proposed in [5], which uses the mesh network topology of WirelessHART to apply BFS to select neighbours. ELFHR takes advantage of the fact that the WirelessHART network manager has enough resources to create routing paths, using least-hop as a metric for all network nodes. Thus, ELFHR does not have to search for the quickest paths to all network nodes, instead focusing only on those specific nodes considered branches of the breadth-first tree.

IWSN lifetimes are shown to be adversely impacted by irregular and early energy exhaustion of separate network nodes. Therefore, previous solutions for effective energy balanced network routing included the implementation of several sinks linked through either wireless or interconnected structures to balance the energy usage of nodes. The authors in [12], propose an Energy-Balancing Routing algorithm based on Energy Consumption (EBREC) to address this, where nodes select others to communicate with based on their possible function, but this does lead to a delay in network communication. The proposal is compliant with existing WirelessHART standards and can therefore be easily implemented into current wireless systems; however, the authors did not consider other factors that could also result in energy depletion in WirelessHART networks, such as conflicts during network communications.

Similarly, the authors in [17] stated that the effectiveness of WirelessHART in industrial settings is often constrained by energy levels and communication requirements. Previous recommendations included the use of the minimum transmission power cooperative routing algorithm, which lessens energy usage for a specific network route while providing minimum throughput guarantees. However, this algorithm does not consider the closest nodes’ remaining energy and transmission capacity, under high traffic load.

According to [9], several industrial standards such as WirelessHART use time slotted channel hopping, which is a TDMA network topology to ensure consistent communication. However, network communications can be delayed in such cases by conflicts during transmissions on a general device. Previously proposed solutions included the use of a local set of rules that proactively chooses the next hop by which to send a packet; however, these authors argued that the proposed decentralised methods do not offer end-to-end delay assurances and thus cannot be implemented with the latest standards for IWSN. They instead suggest a conflict-aware routing protocol, which is a traditional method for real-time routing in WirelessHART networks. This approach integrates communication conflicts and planning with routing decisions to enhance real-time communications.

Recently, different reinforcement learning models in the WSNs were used for data delivery, energy consumption, and latency optimisation. In one of these models, known as Q-Routing, the network nodes learn which of their neighbors delivers the best routes for a destination node. However, this model cannot be used for centralised networks because it does not provide path redundancy, and the nodes cannot select the routes. The Q-Learning Reliable Routing with Multiple Agents (QLRR-MA) approach is presented in [18], which builds routing graphs in a centralised way using the Q-Routing model. The approach results demonstrated that in a significant number of cases, average network latency is reduced.

Similarly, [19] offers the graph routing algorithm Q-learning Graph Routing Lifetime Enhanced (QGRLE) to improve lifetime, latency, and reliability performance IWSNs metrics. The proposed algorithm periodically reconstructs the routing graph while reliability metrics, lifetime, and latency experience dynamic optimisation. As for the power resources, they are fully used considering the nodes’ residual energy. After conducting simulations, the QGRLE algorithm was effective and improved lifetime and latency performance.

The multipath routing (MPR) algorithm is proposed in [20] that provides the industrial wireless mesh networks with low-cost planning, high reliability, and low-level latency. The algorithm builds three primary paths, each consisting of multiple nodes with different hops’ numbers. While the multipath routing algorithm prioritises the data transmission over the shortest path, the alternative paths are always ready to tolerate the transmission errors. The multipath routing algorithm was simulated according to three existing algorithms, including [18]. The MPR algorithm demonstrated results that the significant reduction in average network latency, enhancement of expected network lifetime, and the ratio of data packet deliver.

However, previous solutions have typically focused on the suggestion of network routes based on traditional route optimisation methods, with the exception of two works [18,19] that suggested using Q-learning with graph routing of IWSNs. Therefore, the possibility remains open of applying a further stage of optimisation to IWSN GR routing to compute best paths that balance all requirements. The current work uses this strategy to bridge the gap between communication reliability, reducing end-to-end transmission time and balancing energy consumption to increase the lifetime of the network.

### 2.2. Optimisation Techniques Applied to Routing

In recent years, there has been growing interest in applying optimisation techniques to route algorithms to determine the best path. These techniques achieve the best paths by minimising the hop count, in addition to maximising transmission rates or minimising propagation or queuing delays. In this section, we present some state-of-the-art research that employed optimisation techniques to improve the performance of the routing algorithms. A survey on optimisation techniques for WSNs was recently conducted in [11], although the routing algorithms described are not related to IWSNs.

To achieve best paths, the authors in [21] combined Ant Colony Optimisation (ACO) with a minimum hop count scheme. ACO determines the best path for routing the data packets in a WSN by using the number of hop counts originating from the source node to the sink node. The routing algorithm used in [21] has optimisation capabilities that determine the shortest path, which in turn, minimises time delays and reduces energy consumption. Meanwhile, [22] uses Ticket-Based Routing (TBR) in WSNs in smart grids to ensure that the forwarding of packets is effective. However, a significant limitation of TBR is that it suffers major setbacks during best routing path discovery. Therefore, to address this, the authors combined TBR and genetic algorithms, which reduces the number of tickets and delays.

To solve the problem of uneven energy consumption, the authors in [23] use genetic algorithms and fruit fly algorithms. These algorithms are applied to cluster the nodes in the network, while the Dijkstra algorithm [24] is used to determine the specific best path. This combination of algorithms optimises the whole network, improving the network lifetime by 50% and boosting the whole network coverage by 10%. The authors in [25] demonstrate how the Tunicate Swarm Grey Wolf Optimisation (TSGWO) algorithm, assisted by IoT agents, can be deployed in WSNs to find a best routing solution. Multipath routing protocols in a network are used to transmit data simultaneously from a single source or node to multiple destinations. Such protocols are typically aided by IoT assisted agents, particularly in WSNs. In [25], the IoT agent-assisted-TSGWO algorithm considers factors such as link time, delays, energy and distance to discover the shortest path in routing. The authors suggested a routing algorithm [26], that combines the Butterfly Optimisation Algorithm (BOA) with ACO in WSNs. The authors argue that this combination improves throughput and reduces energy consumption by means of clustering to achieve overhead routing. Consequently, network lifetime and performance are optimised during implementation. While [27] proposes Ad hoc demand Multipath Distance Vector routing with an Adaptive Grey Wolf optimization algorithm (AOMDV_AGWO) to improve the energy efficiency of WSNs. AOMDV_AGWO applied a genetic algorithm to clustering with ad hoc on-demand multipath distance vector routing with Adaptive Grey Wolf Optimization (AGWO). Where the AGWO technique predicts the optimal path, made from ad hoc on-demand multipath distance vector routing protocol. This technique is used to provide efficient shortest-path communication.

### 2.3. CMA-ES

Drawing upon the principles of biological evolution, the algorithm of the Covariance Matrix Adaptation Evolution Strategy (CMA-ES) [15] uses continuous stochastic search methods. Usually, CMA-ES adopts a multivariate normal mutation distribution method to revise the covariance matrix of variables to achieve the objective function (f). There is similarity between the performance of this algorithm and the reverse matrix in the Newton method [28], however, this algorithm does not require gradient analytic computing which has difficulty finding the best solution due to lack of differentiability [16].

To solve problems of optimisation, the general algorithm samples several independent points from a given distribution, P. These points are appraised according to their performance, f, from which the distribution parameters are updated. This process continues until the termination criterion is achieved. In the CMA-ES algorithm, P is a multivariate normal distribution that is a generalisation of the one dimensional (univariate) normal distribution to higher dimensions. Consequently, if every linear combination of a vector’s n components have a univariate normal distribution, the distribution of the random vector is regarded to be n-variate normal. The entropy of the mean values, variances and covariances make normal distribution a suitable candidate for randomised searches because the distributions in ${ℜ}^{n}$ are the largest, and there is no differentiation between the coordinate directions. Therefore, CMA-ES samples a multivariate normal distribution to create a population of new search points (set of individuals). In every iteration, g new individuals $x_{i}^{g}\in {ℜ}^{n}$ are calculated as:(1)$$x_{i}^{g+1}=m^{g}+\sigma^{g}\times N_{i}^{g}\left(0,C^{g}\right)\;i=1,\ldots ,\lambda \;\left(1\right)$$
where $m^{g}$ denotes the approximated mean value, $\sigma^{g}\;$> 0 is the standard deviation—step-size at the *g*^th^ iteration, $N_{i}^{g}\left(0,C^{g}\right)$ is a normal distribution with the mean 0 and λ is a population size, $C^{g}$ the n×n covariance matrix of the search distribution. Therefore, a mutation arises randomly when the covariance matrix is perturbed; resulting in the matrix being iteratively revised, prompting a search for areas in which the objective values are anticipated to be lower. Once a population of individuals are generated, f is used to assess them, then they are sorted and transformed as described in (1). All distribution parameters $(m^{g},$ $C^{g},$ $\sigma^{g})$ are revised after each iteration. Further details of CMA-ES algorithm can be found in [15].

## 3. Description the Model of Best Paths for Graph Routing Based on CMA-ES Selection

### 3.1. Overview

This work focuses on how sensor nodes in IWSN monitoring systems create the uplink graph that they use when transmitting sensor data to the $G_{w}$, where details of the GR algorithm mechanism are in [8]. Mesh topologies were selected because IWSNs are commonly of this type, with static sensor nodes powered by batteries [2]. The network is thus assumed to operate with this topology during the simulations. Nodes are also assumed to inform the NM about poor connections with neighbours, so that the NM can remove these connections from the network topology. Our GR algorithm was then evaluated against the three requirements of IWSN applications: high communication reliability, balanced energy consumption, and low end-to-end transmission time [2,4]. In order to evaluate these requirements, the following metrics have been defined.

The average *EIF* is defined as the standard variance of the residual energy of all nodes in the network. It is used to demonstrate how effective a GR algorithm is in terms of achieving an energy balance. The *EIF* is calculated as
(2)$$EIF=\left(\;\frac{1}{n}\sqrt{\overset{n}{\underset{i=1}{\sum }}{\left(RE_{i}-RE_{avg}\right)}^{2}\;}\right)$$
where n is the number of nodes, $RE_{i}$ is node i’s residual energy, and *RE_avg_* is the average residual energy of all nodes.
Reliability is evaluated using PDR, which is the ratio of data packets that successfully reach the *G_w_* to the total number of data packets sent by the source nodes.In addition, the PMR is defined as the ratio of data packets that failed to make it to the *G_w_*The latency of each data packet is evaluated using E2ET, which is the time required for a data packet to travel from the source node to the $G_{w}$

### 3.2. Covariance Matrix Adaptation Evolution Strategy (CMA-ES) for Graph Routing in IWSN

CMA-ES, introduced in [15], is an effective evolutionary algorithm for global optimisation problems by finding the optimiser, x, of a real-valued objective function f. CMA-ES employs two evolution paths to realise the necessary exploitation and exploration during the search process, which are the updating of the covariance matrix and the learning of the covariance matrix.

This article’s model selects the best path for graph routing in a WirelessHART network based on CMA-ES, as presented in Figure 1, which portrays a schematic view of the proposed routing model. The main operations include generating sampling, evaluation of the best path based on three objective functions $f_{D}$, $f_{E}$ and $f_{E2ET}$, covariance matrix adaptation, path evolution and global step size adaptation and the output of best single-objective paths, PODis, POEng and POE2E. The final step is the adaptation of CMA-ES to select the final best path for Graph Routing with multiple-objectives (BPGR-ES).

#### 3.2.1. Objective Functions of the Best Path

The objective functions are used in our model to select the best path for each source node in the WirelessHART network. To achieve the first objective which is called $f_{D}$, the Euclidean distance between the sensor nodes and their neighbours heading in the $G_{w}$ direction is calculated. Subsequently, this is utilised to select the best next hop to minimise transmission distance [29]. The second objective function is the residual energy for each node. Which is used to avoid dead nodes and to minimise the energy consumption. This is known as $f_{E}$. The third objective function is $f_{E2ET}$ where the end-to-end transmission time is then considered in the selection of the best next hop as a way to reduce conflict delay. The pseudo code of the three single objective functions of CMA-ES, considered to select the best GR path, is shown in Algorithm 1 and discussed in greater detail in the following.

Minimum communication distance between the source node and receiver node toward $G_{w}$, $\left(f_{D}\right)$: This is defined as the minimum distance between the currently sending node and its neighbours in the $G_{w}$ direction and achieved by minimising D, the currently sending node, with the lowest communication cost. Thus,
(3)$$f_{D}=Min\left(D_{PathFromnode}{}_{i,j}\right)$$
where $D_{PathFromnode}{}_{i,j}$ is the Euclidean distance between the currently sending node i and its neighbour node j toward the $G_{w}$ in Shortlist and is used in this model rather than the neighbouring table to build all the best paths.


After the deployment of sensor nodes in the WirelessHART network area, each sensor node submits its neighbour table to the Data Link Layer (DLL) in the NM. The routing formation prior to the transfer of data at the Network Layer (NL) can therefore use this neighbour table and a connected graph to construct a routing table for each sensor node. As wireless conditions change frequently in industrial environments, the NM must also frequently reconfigure and re-disseminate the routing graphs, which leads to increased energy and bandwidth consumption. To address this, as shown in the pseudo code in Algorithm 2, this model saves the storage space required and reduces the large overhead for the routing table by allowing the source sensor to select its neighbouring sensor node closest to the
$G_{w}$, being the Shortlist, while the neighbouring table for each source node retains all its neighbours in each direction in the network area within an effective communication range. To build the Shortlist in this model for each sensor node in the network (as shown in lines 5–6 of Algorithm 2), the current node reads the neighbouring table at the DLL, then identifies the Euclidean distance between the current node and the $G_{w}$ in line. The current node verifies if the $G_{w}$ is available in the neighbouring table in lines 7–8. This indicates that the current node can communicate directly with the $G_{w}$ as each sensor node has a communication range. From lines 11–15, the current node selects its neighbour nodes for the Shortlist where the distance between neighbour node of the current node and $G_{w}$ is less than distance between the current node and $G_{w}$.
**Algorithm 1:** Objective Functions of Select the Best Paths of Graph Routing algorithms based on of CMA-ES.1:**Input:**2:   Shortlist of Source Sensor Node3:**Output:**4:   Optimal paths based on CMAES algorithm5:$For\;Shortlist_{Distance}$6:  $\;\mathrm{Calculate}\;Distance\;between\;\left(Final\;destination,\;Node\left(Shortlist_{Distance}\right)\right)$7:   If Distance<minD 8:         $SelectedNodeD=Shortlist_{Distance};$9:         Find MinD=Distance;10:   **End**11:**End**12:$ForShortlist_{Energy}$13:   $If\;Node\left(Shortlist_{Energy}\right).CurrentEnergy>=MaxE$14:       $SelectedNodeE=Shortlist_{Energy};$15:      $\;\mathrm{Find}\;MaxE=Node\left(Shortlist_{Energy}\right).CurrentEnergy;$16:   **End**17:**End**18:$ForShortlist_{E2ET}$19:   Delta=FindDeltaFromPropagationModel();20:   $\;\mathrm{Calculate}\;Distance\;between\;\left(Final\;destination,\;Node\;\left(Shortlist_{Distance}\right)\right)$21:    Calculate E2ET=(c1*nodeprocessor)+(c2*d*delta);22:      If E2ET<=minDelay23:         $SelectedNodeDelay=Shortlist_{E2ET};$24:         Find MinDelay=E2ET;25:      **End**26:**End**



**Algorithm 2:** Build Shortlist of Sensor Nodes.
1:
**Input:**
2:      *Source Node*3:
**Output:**
4:      *Shortlist baesd on Neighbouring Table*5:      *CurrentHop = Source Node*6:      *CurrentHop read neigbouring Table*7:  **For**8:  ***If** Node*(*CurrentHup*).*neighbouringTable* == *G_w_*9:   Calculate *Distance between* (*Node*(*CurrentHop*), *G_w_*);10:     **For**11:      *Node*(*CurrentHop*).*neighbouringTable*;12:       Calculate *Distance_Hup between*(*Node*(*CurrentHop*), *Node* (*DNode*))13:
*        **If** Distant_Hup < Distant*
14:         Add *Node* (*CurrentHop*) *to Shortlist*15:        **End**16:     **End**17:  **End**18:
**End**





2.Maximum Residual Energy: This is defined as the residual energy in the sensor nodes after they perform sensing, communication operations and computation. Sensor nodes with higher residual energy tend to be selected as the next hop in the best path, as maximising $f_{E}$, each sensor node periodically uploads its residual energy to NM. Thus,
(4)$$f_{E}=Max\;\left(E_{Current}{}_{i}\right)$$
where $E_{Current}{}_{i}$ is the residual energy of sensor node i. To ensure the quality of communication and increase reliability in the IWSNs, each currently sending node looks in Shortlist N to locate the sensor node with maximum energy in the required path rather than examining the neighbouring table.3.End-to-End transmission time between the source node and receiver node toward the $G_{w},$ $\left(f_{E2ET}\right)$: The End-to-End transmission time measure proposed in this research refers to the time required for a given pair of nodes in the WirelessHART network to exchange a data packet. WirelessHART is a TDMA-based network protocol. Each communication is time-synchronised and this provides a timescale for nodes in the network. A fixed-length timeslot shared by all network devices is the basic time unit of communication activity. Seeing that all hardware clocks are imperfect, those at different nodes may drift away from each other. For this reason, the observed time or time interval durations may differ for each node in the network. The timeslot hence provides a time base for scheduling the transmission of process data. In WirelessHART, a timeslot has a duration of 10 ms, which is sufficient to send or receive one packet per channel and its accompanying acknowledgement, including the guard-band times required for network-wide synchronisation.


Several mechanisms are applied in wireless networks for time synchronisation. In this research, in order to obtain a definitive means of identifying the time required for data packet exchange between any two sensor nodes in the WirelessHART network, a Two-way Time Message Exchange (TTME) clock offset estimation model was applied between each pair of nodes in the sensor network, as in [30]. This allowed the development of an equation that simulates the actual transmission time for each packet between the transmitter and the receiver based on the propagation model in a WirelessHART network [7], as shown in Figure 2. The uplink-downlink time was modelled using the following equation:
(5)$$T_{i,j}=c1\tau_{i}+c2D_{ij}\partial_{l}$$
where c1∈[0,1] and c2∈[0,1] are the node’s processing delay time and channel delay time coefficients, respectively, $\tau_{i}$ is the processing time required by node i to process a data packet, $D_{ij}$ is the distance between the transmitter node i and the receiver node j and $\partial_{l}\in \left[0,1\right]$ is the delay time required to transfer a data packet from the transmitter node i and the receiver node j through channel l.
(6)$$f_{E2ET}=Min\left(T_{i,j}\right)$$
where the $Min\left(T_{i,j}\right)$ of $f_{E2ET}$, is the minimum time required from source node i to receiver node j in the Shortlist N.

At the end of the CMA-ES, there are three best GR paths, specifically PODis, POEng and POE2E. Finally, to select the final best BPGR-ES path and achieve a balance between the three objectives above and energy consumption between sensor nodes in the network, we adapt the CMA-ES to select the final best path based on three objectives by means of Algorithm 3, which is discussed in greater detail in the next section.


**Algorithm 3:** Selection Best Path of Graph Routing (BPGR-ES).
1:
**Input:**
2:    *PODis*; *POEng*; *POE2E*3:
**Output:**
4:    *Final Best Path* (BPGR-ES)5:***If** isequal* ((*PODis*, *POEng*) && (*POEng*, *POE2E*))6:           *BestPath = POE2E*;7:
**End**
8:   ***If***
*isequal*(*POEng*, *PODis*)9:                      *BestPath = PODis*;10:    ***Else if***
*isequal*(*PODis*, *POE2E*)11:                      *BestPath = POE2E*;12:    ***Else if***
*isequal*(*POEng*, *POE2E*)13:                      *BestPath = POEng*14:   **End**15:**If***isnotequal* ((*PODis*, *POEng*) && (*POEng*, *POE2E*))16:    *Check number of the hops for all the paths*17:   *Select BestPath which has smaller number of the hops*18:    ***If** all paths OR two paths have the same number of the hops*19:     *Check energy of last node before G_w_ in these paths*20:      *Select BestPath which has heigh energy for last node before the G_w_*21:    **End**22:
**End**




#### 3.2.2. CMA-ES Adaptation for BPGR-ES Final Best Path Selection

To meet the final best path selection criterion in an actual WirelessHART network, data packets are forwarded via the final best path, as shown in Algorithm 3. In relation to the pseudo-code for the selection of the final best BPGR-ES paths, there are two situations in which the final best path must be selected, namely equality or inequality, which apply to all best paths, based on PODis, POEng and POE2E.

In the case of equality, as shown in lines 6–8 in Algorithm 3, there are two cases: if all objectives of the objective functions for transmission are achieved in one path, this will be the best path. Hence, this must be selected as the final best path. In the second case in lines 8–14, if there are two potential best paths that have the same path, priority will be given to any one of them as the final best path where two objectives of the objective functions are achieved. In a situation where there is inequality between the best paths, as observed in lines 15–21, the final best path with the least number of hops will be selected to reduce energy consumption and increase reliability. However, as a number of best paths may have the same number of hops, a further check on the residual energy of the last sensor node around the $G_{w}$ is added to achieve balanced energy consumption. Subsequently, priority is given to the best path which has the highest residual energy at the last sensor node before the $G_{w}$.

## 4. Simulation Experiments

### 4.1. Simulation Setup

Simulations were conducted using MATLAB R2020b over a Windows 10 workstation running on an Intel (R) core™ i7 processor with 16 GB RAM. The uplink GR algorithm was applied to a mesh topology model for the WirelessHART network and randomly generated the sensor nodes, as shown in Figure 3. The topology consisted of one red rhombus, which is the $G_{w}$ positioned in the centre of the network area; two blue squares are the APs, located 10 m to the right and 10 m to the left of the $G_{w}$; along with 50 wireless sensor nodes placed at random [31]. The connections between the $G_{w}$ and the various APs were considered to be reliable and wired. The sensor nodes were battery-powered and stationary after deployment, as this is common in real industrial environments [32]. Each sensor node was also assigned a unique ID. For any WirelessHART network, the maximum packet size sent should be 133 bytes [2]. Each node in a system was assumed to be homogeneous in terms of having the same size and energy. The maximum energy of each node was thus assumed to be 0.5 J [8]. The energy model described in [8] As real IWSN are subject to different wireless channel conditions, a general path loss model for Received Signal Strength Indicator (RSSI) estimation was incorporated. A packet loss physical layer probability model, and O-QPSK modulation were also included in the propagation model [7] based on the WirelessHART standard. A TDMA scheduling scheme was also employed in the simulation, with all simulations utilising the same scheme to allow for fair comparisons. Table 1 shows the system parameters.

The CMA-ES algorithm does not require tuning of the parameters with the exception of population size *λ*, where strategy parameters are considered a part of the algorithm design. This is a feature of CMA-ES. The aim is to have a well-performing algorithm as observed in [15]. Therefore, we set λ=4+⌊3 log(n)⌋ as suggested in [15] where *n* is the number of the variables that are in the Shortlist in this model. The parameter σ specifies the direction of the algorithm was considered as 0.3 × (*VarMax* − *VarMin*), where VarMax, VarMin are upper and lower bound to the Shortlist decision, respectively. Each simulation begins with the initialisation of the NM, $G_{w}$ and APs. The NM then builds the configurations for the relevant network (routes and links), based on its knowledge of each node in the network, including its location and its battery status. This data is derived from the health reports sent by the sensor nodes every 15 min. When a new node joins the network, it receives network configurations from the NM after each update. Each simulation was run for 4 h. Table 2 shows the CMA-ES parameters.

To study how the PODis, POEng, POE2E and BPGR-ES perform under various network sizes and different numbers of sensor nodes, extensive simulations were conducted to evaluate their performance in four scenarios as compared with the baseline uplink algorithms in [5] and [12], specifically the Enhanced Least-Hop First Routing (ELHFR) algorithm and the Energy-Balancing Routing algorithm based on Energy Consumption (EBREC).

Transmission power was first set to 0 dBm and a maximum communication range of 35 m for a 100 × 100 $m^{2}$ network area. The transmission power was then increased to 10 dBm with a maximum communication range of 75 m, permitting a 200 × 200 $m^{2}$ network area [7]. In each case, 50 or 100 sensor nodes were used to verify the algorithms’ performance under varying node densities [33]. As each run of the simulation presented a different node topology with respect to the spatial distribution of sensor nodes, the performance metrics generated were for different values. Several simulations were therefore conducted to verify whether algorithms produced similar performance levels over 15 random topologies in order to obtain the statistical mean for the results. The total consumed energy, average EIF, PDR, PMR and E2ET results for several repetitions of the simulation for each algorithm were obtained.

### 4.2. Evaluation Results and Analysis

#### 4.2.1. Network Reliability Evaluation

The two critical factors used to evaluate the reliability of the network in this research are PDR and PMR, as shown in Figure 4. With an increase in the delivery of data packets to the $G_{w}$, the packet miss ratio decreased.

In both the 50-node and the 100-node topologies across the two network sizes, BPGR-ES presented the highest PDR and lowest PMR. As shown in Figure 4, as the number of sensor nodes increased, the PDR decreased very little across most algorithms. This is reasonable, because, as the number of sensor nodes increased, the traffic load in the network intensified, causing congestion in some areas and data packet loss throughout the network. As a result, the PDR was negatively affected. Nonetheless, as Figure 4 demonstrates, the proposed BPGR-ES approach still attained the highest PDR and lowest PMR as compared to other options, even with such increases in the number of sensor nodes across different network sizes. This is because BPGR-ES allows communication with all neighbours in the $G_{w}$ direction that are on the Shortlist regardless of whether these are at the same level or lower levels, according to Euclidean distance. Therefore, this increases the availability of sensor nodes and, hence, reduces the PMR and increases the PDR. This also observed in all single-objective best paths, which also used the Shortlist.

Furthermore, BPGR-ES selects the neighbour used in the next hop from the Shortlist to help ensure delivery of the packet within its deadline, as well as achieving the most reliable possible paths through retransmission of data packets where the next-hop step is unable to receive the data packet. The number of lost packets due to path redundancy is, therefore, reduced, improving network throughput. Taking these important parameters into account explains the observed reduction in the packet miss ratio. BPGR-ES presented similar PDRs for the 50-node topology throughout various network sizes, although it exhibited a slight decline in PDR of approximately 1.01% in the 100-node topology of the larger network size. Generally, all best POE2E, POEng and PODis paths exhibited good PDR results with a maximum of 98.94% and a minimum of 98.67%, while the PMR was approximately 1.43% as a maximum of the POEng (see Figure 4b). The ELHFR algorithms produced lower PDR results of approximately 1.93% for the 50-node topology in both network sizes. Nevertheless, since this only permits sensor nodes to establish connections with neighbours located at lower levels in the BFS tree, fewer neighbours are typically available in the lower levels [5]. In denser networks, ELHFR reduces the PDR by approximately 1.95% because it does not guarantee path redundancy for every sensor node while increasing the PMR.

#### 4.2.2. Energy Consumption Evaluation

The performance of the proposed BPGR-ES approach was evaluated with respect to energy consumption in terms of both total consumed energy and average EIF of the energy balance. This is important, as the IWSNs are centralised, making balancing energy consumption between sensor nodes a key target [4].

As shown in Figure 5, the BPGR-ES algorithm reduced the total energy consumption of the 50-node topology across different network sizes in contrast to the other algorithms. However, in the denser networks, the total consumed energy appeared extremely similar between the proposed BPGR-ES approach and EBREC. This is due to the fact that the EBREC algorithm considers the remaining energy when communicating with the nodes in the network. Nevertheless, the BPGR-ES approach fares better concerning total consumed energy because, as shown in Figure 4, the network connection is better, with 99.77–98.78% data packet delivery.

Notably, upon increasing sensor nodes, total energy consumption of PODis is evidently reduced due to the larger number of sensor nodes nodes, with the shortest path being chosen in the quickest way, thereby significantly reducing energy consumption. This stems from reliance upon the Shortlist defining the neighbours per sensor node with the least Euclidean distance to the $G_{w}$. Yet, this significantly increases the network sensor nodes’ energy consumption imbalance, (see Figure 6). Thus, the PODis fails to balance energy consumption. It should be noted that this is logical because the energy consumption of the sensor nodes nearest the $G_{w}$ increases due to its constant selection of the shortest path depending on the Euclidean distance from the source nodes farther from the $G_{w}$. In POEng, total consumed energy sharply increased for all different network sizes compared to other algorithms, as shown in Figure 5. Moreover, because it selects neighbouring sensor nodes with the highest residual energy from a Shortlist without taking into account the distance from the $G_{w}$, this increases the number of hops. This causes increased energy consumption compared to other algorithms, where proximity to the $G_{w}$ is not considered to reduce the number of hops noticed while the simulation is running. Typically, the number of the hops in POEng is more than other paths. Consequently, as shown in Figure 6, the high energy consumption of the POEng evidently affects energy consumption imbalance among the network nodes.

The average EIF of the proposed BPGR-ES approach was also significant being the smallest among those tested according to the results in Figure 6. The high use of the sensor nodes around the $G_{w}$ as compared to other nodes resulted in a reduction in the average residual energy, which led to an increase in the average EIF. This suggests that in the BPGR-ES algorithm, the energy of all the nodes in the network is closer to the average energy than in the other approaches, as the BPGR-ES approach selects the best path. This is based on the highest remaining energy of the sensor node around the $G_{w}$ only where all best paths have the same number of hops. Therefore, the proposed BPGR-ES algorithm achieves a better balance in terms of energy consumption than other algorithms. However, while all sensor nodes in the EBREC algorithm route their data packets via nodes that have greater residual energy, this is insufficient to ensure a balance in energy consumption between sensor nodes in the network. In particular, if several sensor nodes select the same node to forward their data packets to, this node will take on a critical role, which can lead to imbalances in energy consumption in the network. In addition, the ELHFR algorithm significantly increases average EIF due to its consideration of least-hop as the only selection metric. This does not consider the increased energy consumption of the nodes around the $G_{w}$, as sensor nodes closer to the $G_{w}$ become overburdened with high traffic loads compared to those further away. Therefore, these overloaded nodes will expire much faster than the other sensor nodes due to such imbalances in energy consumption.

#### 4.2.3. End-to-End Transmission Time Evaluation

A further experiment examined the proposed approaches in terms of end-to-end transmission time (E2ET). Monitoring systems often have delay needs of fewer than 100 ms, whereas factory automation has even stricter delay requirements ranging from 2 to 25 ms [34].

Figure 7 shows the various E2ET results for the network topologies for a 10,000-round run of each algorithm. The simulation results clearly show that the E2ET results of all algorithms increased with the 100-node topology across different network sizes compared to the 50-node topology, where the increased network traffic prompted a rise in the nodes’ multi-hop behaviours and path redundancy. Consequently, an increase in the E2ET occurred due to several data packet retransmissions from the sensor node to the $G_{w}$, which caused queuing and other delays.

It is nevertheless evident that the POE2E of GR gave the lowest E2ET results compared with all other topologies, where the highest transmission time of POE2E in a 100 × 100 m^2^ network area reached 11 ms, while that in the 200 × 200 m^2^ network area reached 17 ms, as shown in Figure 7. This is justifiable for the following reasons: the POE2E synchronises the time required to select the data packet transmission time between a source node and receiver node to form the next hop in the best path and improves the packet delivery by preventing the use of unreliable paths. All delays due to retransmission of lost packets are, therefore, reduced. Moreover, the POE2E decreases the congestion at the nodes by selecting the neighbours that can best deliver the data packet within its deadline. This is achieved by using a Shortlist to choose the next hop from the available neighbours, hence facilitating faster delivery of data packets and reducing the delay. Even if transmission of the network data packets is broken, the intermediate nodes will not spend any time searching for the next hop for the retransmission of data packets, which accelerates packet forwarding procedures in terms of reaching the $G_{w}$. This was also observed in the E2ET results of the PODis, where transmission times reached 15 ms and 21 ms, as shown in Figure 7, with 50 and 100 sensor nodes, respectively. Some delays occurred in the E2ET results of the PODis approach compared with POE2ET because the PODis approach selects best paths as the shortest paths, increasing network traffic, especially of sensor nodes around the $G_{w}$, and prompting delays. In addition, the BPGR-ES approach selects best paths as in the POE2ET approach in one case if this is the best path between all of the best single-objective paths but prioritises the balance of energy consumption in the other cases. Thus, as shown in Figure 7, compared to the POE2ET approach, transmission time increased up to 21 ms for different network topologies of BPGR-ES. 

Figure 7 illustrates a significantly sharp increase in the E2ET results of POEng, with the transmission time reaching 55 ms. The main reason for this is the increased number of hops, which caused data packets to arrive at the $G_{w}$ with a noticeable delay. In the ELHFR and EBREC algorithms, however, data packets could not avoid heavily congested regions. Therefore, it may have taken a long time for them to look up options in the neighbouring table to locate the next hop node in the path. Consequently, this increased the transmission time due to the retransmission of multiple data packets striving to reach the $G_{w}$.

### 4.3. Performance Comparison 

Table 3 presents the performance comparison based on the results in Section 4.2, comparing the proposed POE2ET, POEng, PODis, and BPGR-ES approaches with the state-of-the-art GR algorithms. In terms of the following items, their criteria of paths, reliability, balance of energy consumption, and transmission time. The following list of items will be discussed:Criteria of paths: the primary path and formula specified by the GR algorithm for each sensor node (i.e., which is the path by which a sensor node will attempt to send a data packet for the first time) and what is the criterion for this selection;Reliability: the ratio of delivery of the data packets to the $G_{w}$ measured by averaging the PDR results for each algorithm across the different topologies;Balance of energy consumption: the ration of energy consumption balance between all of sensor nodes in the network area determined by averaging the EIF results for each algorithm across the various topologies;Transmission time: the time it takes each algorithm to send a data packet from a source sensor node to the $G_{w}$ determined by lower time and higher time in the E2ET results for each algorithm across the different topologies.


## 5. Conclusions and Future Work

This research adopts a Covariance-Matrix Adaptation Evolution Strategy (CMA-ES) to establish best paths of a graph routing algorithm for Industrial Wireless Sensor Networks (IWSNs) that also provide path redundancy. Firstly, this research proposed three best paths, each based on a single-objective function for CMA-ES according to the different performance requirements of IWSNs considered in this research: the best Path based on the Distance between sensor nodes in the direction of the gateway (PODis), the best Path based on residual Energy (POEng) and the best Path based on the End-to-End transmission time (POE2E). Secondly, this research proposes the best Path of Graph Routing-Evolution Strategy (BPGR-ES) algorithm, which selects the best hops on the basis of multiple objectives to achieve balanced energy consumption as well as a balance among IWSN requirements. This research has evaluated the three best single-objective paths (PODis, POEng and POE2E) and the best path with multiple objectives (BPGR-ES) across several different topologies in order to examine the total consumed energy, End-to-End Transmission (E2ET), average Energy Imbalance Factor (EIF), Packet Delivery Ratio (PDR) and Packet Miss Ratio (PMR).

The results revealed a reduction in E2ET across all topologies for the POE2ET algorithm. Additionally, the PDR values were good for all proposed approaches: 99.57%, 98.84%, 98.75% and 98.8% for BPGR-ES, PODis, POEng and POE2E, respectively. Despite the fact that total consumed energy for PODis outperformed BPGR-ES in small networks and that total consumed energy for BPGR-ES and EBREC was somewhat similar in dense networks, the BPGR-ES algorithm achieved an 87.73% better energy balance among all sensor nodes in the network in terms of average EIF. It is also noteworthy that all the best single-objective paths of GR did not achieve balanced energy consumption over a mesh topology. It is also noteworthy that all the best single-objective paths of GR did not achieve balanced energy consumption over a mesh topology. Future work will, therefore, strive to implement the best single-objective paths of GR with unequal clustering topology to evaluate its performance. Lastly, research directions to explore include the development of the GR to build the best paths on the basis of mobility, scheduling, and real-time requirements, simulation experiments using other IWSN standards, and the evaluation of several state-of-the-art graph routing algorithms in real IWSN deployments.

## Figures and Tables

**Figure 1 sensors-22-07462-f001:**
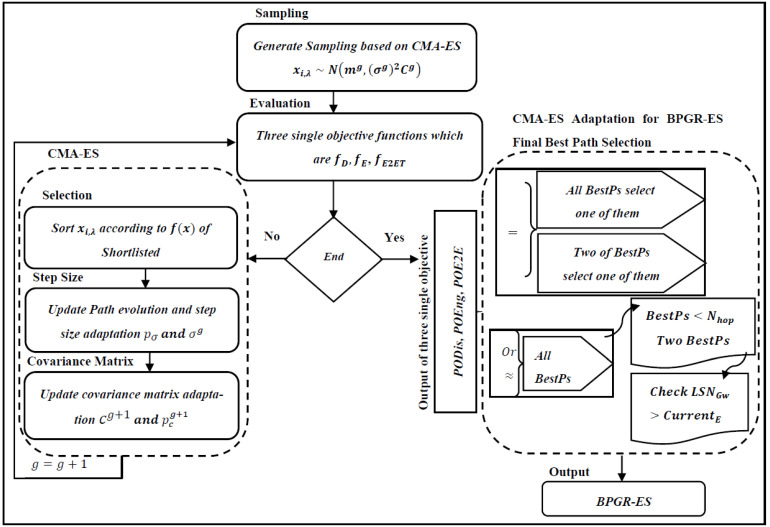
Schematic view of proposed Graph Routing model.

**Figure 2 sensors-22-07462-f002:**
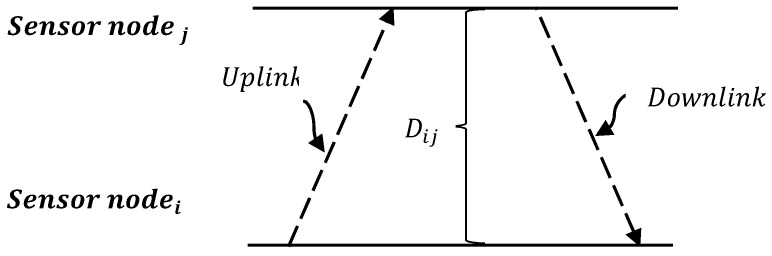
Two-way time message exchange between node *i* and node *j*.

**Figure 3 sensors-22-07462-f003:**
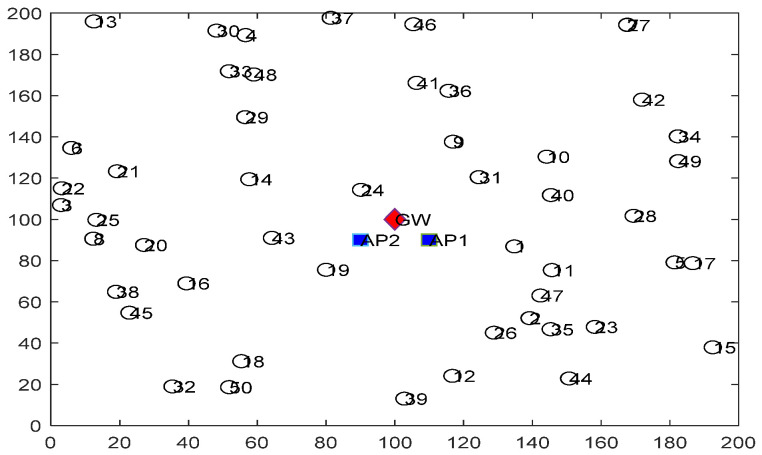
A network topology example with 50 sensor nodes.

**Figure 4 sensors-22-07462-f004:**
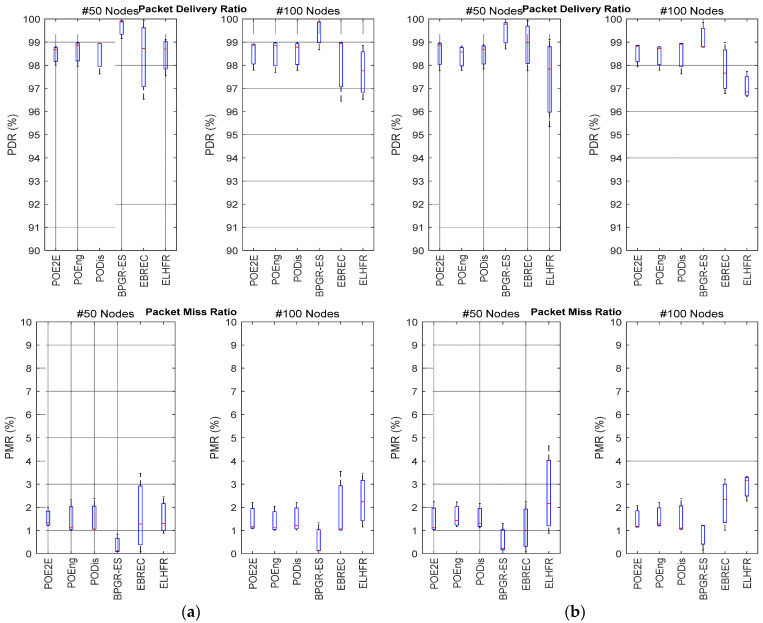
PDR and PMR boxplots for different topologies: (**a**) PDR and PMR results of the 100 × 100 m^2^ network area of 50 and 100 nodes; (**b**) PDR and PMR results of the 200 × 200 m^2^ network area of 50 and 100 nodes.

**Figure 5 sensors-22-07462-f005:**
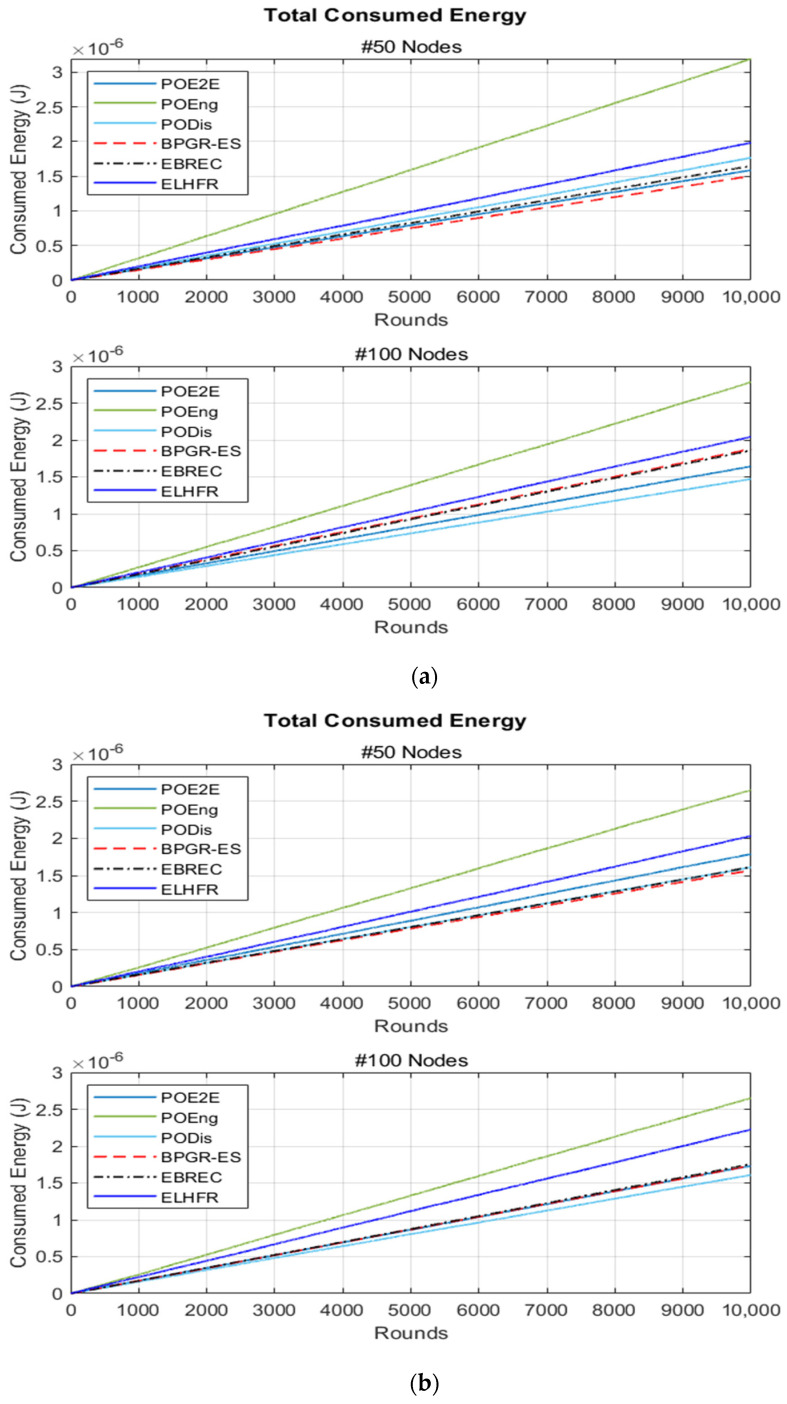
Total Consumed Energy for different topologies: (**a**) energy consumption results of the 100 × 100 m^2^ network area of 50 and 100 sensor nodes; (**b**) energy consumption results of the 200 × 200 m^2^ network area of 50 and 100 sensor nodes.

**Figure 6 sensors-22-07462-f006:**
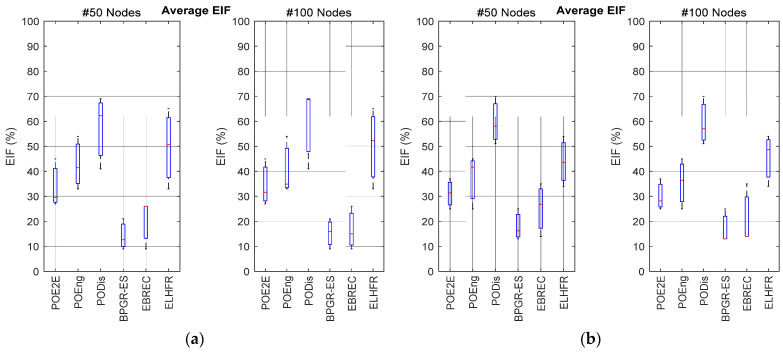
Average of energy imbalance factor for different topologies: (**a**) average EIF results of the 100 × 100 m^2^ network area of 50 and 100 sensor nodes; (**b**) average EIF results of the 200 × 200 m^2^ network area of 50 and 100 sensor nodes.

**Figure 7 sensors-22-07462-f007:**
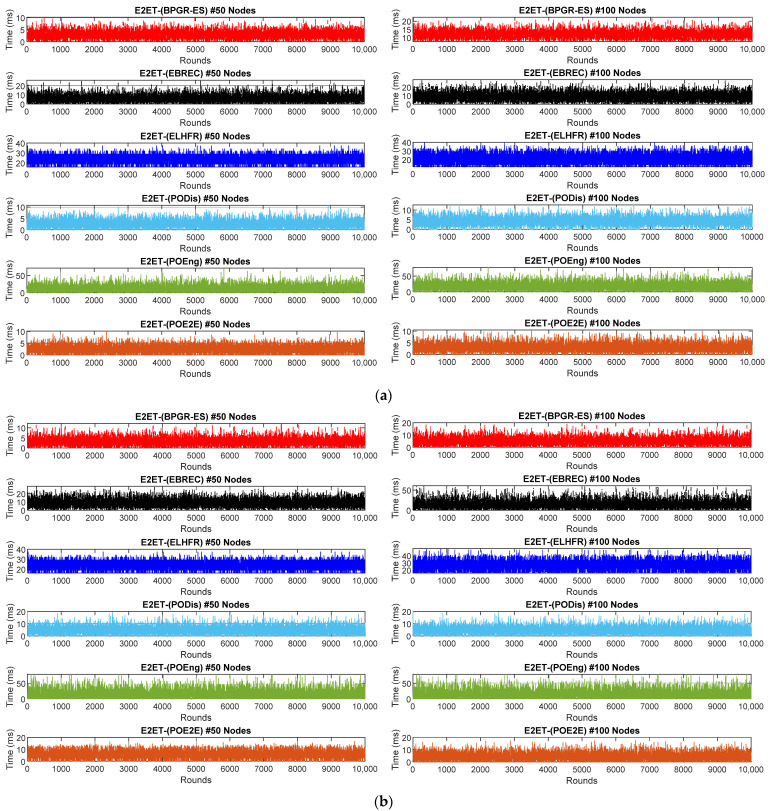
End-to-End transmission time for different topologies: (**a**) E2ET results of 100 × 100 m^2^ network area of 50 and 100 sensor nodes; (**b**) E2ET results of 200 × 200 m^2^ network area of 50 and 100 sensor nodes.

**Table 1 sensors-22-07462-t001:** System Parameters.

Parameters	Value
Simulation area	100 × 100 $m^{2}$ and 200 × 200 $m^{2}$
Number of nodes	50 and 100
Nodes positions	Random
Gateway ($G_{w})$	One $G_{w}$
Access points (APs)	Two APs
Physical layer	IEEE 802.15.4 (2006)
Propagation Model	O-QPSK
Communication range	35 and 75 m
Transmission power	0 dBm
Node initial energy	0.5 J
Maximum Packet size	133 Bytes
Radio frequency	2.4 GHz
Medium Access Control (MAC)	TDMA with 10 ms time slot

**Table 2 sensors-22-07462-t002:** CMA-ES Parameters.

Parameters	Value
Population size (λ)	4+⌊3 log(n)⌋
Number of the variables (n)	Shortlist
Specifies the direction (σ)	0.3×(VarMax−VarMin)
VarMax	Upper bound to the Shortlist decision
VarMin	Lower bound to the Shortlist decision

**Table 3 sensors-22-07462-t003:** Performance comparison of proposed best paths with GR algorithms.

GR Algorithms	Criteria of Paths	Reliability	Balance of Energy Consumption	Transmission Time
POE2ET	Lower transmission time of CMA-ES	98.8%	75.1%	Between 4 to 17 ms
POEng	Highest residual energy of CMA-ES	98.75%	57.88%	Between 7 to 55 ms
PODis	Shortest distance of CMA-ES	98.84%	37.33%	Between 5 to 21 ms
BPGR-ES	Multiple Objectives of CMA-ES	99.57%	87.73%	Between 5 to 25 ms
EBREC [12]	Highest residual energy of BFS	98.6%	86.28%	Between 8 to 53 ms
ELHFR [5]	Highest received signal level of BFS	97.78%	51.2%	Between 7 to 48 ms

## Data Availability

Not applicable.

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
