# Peer review of "Enhancing Graph Routing Algorithm of Industrial Wireless Sensor Networks Using the Covariance-Matrix Adaptation Evolution Strategy"

_sensors, 2022, doi:10.3390/s22197462_

Round 1

Reviewer 1 Report

This paper proposed a routing algorithm for IWSNs. However, the following corrections should be considered:

1- Acronyms should be written in an appropriate way, e.g., "swarm intelligence (SI) -> Swarm Intelligence (SI)." However, some of the acronyms are correct, e.g., Covariance-Matrix Adaptation Evolution Strategy (CMA-ES).

2- The authors should clarify their purpose of optimality and objective function. In computer science and operational research, optimization and objective function are defined in the mathematical formulation.

3- Optimality of the proposed algorithm should be proved.

4- Some figures have very low quality, e.g., Figure 5.

5- The description and goals related to Figure 7 is unclear.

Author Response

Dear reviewers

Thank you for your comments. All of your comments have been taken into account in the revised version of the manuscript.

I've attached the response to reviewer's comments.

Reviewer 2 Report

please find attachment

Author Response

(The authors gave the same response as above.)

Round 2

Reviewer 1 Report

It is so hard to read this version of the manuscript because of the corrections!

As a reader, I can't detect the results that are shown in Figure 7!

Author Response

Dear reviewer

Thank you for your comments. All of your comments have been taken into account in the revised version of the manuscript. 

  • The manuscript has been edited, and all corrections have been removed.
  • The results have been shown in Figure 7 more obviously.
  • The English in the revised manuscript has been edited by MDPI.

I've attached 'English editing certificate'. 

Reviewer 2 Report

Paper is accepted.

Author Response

Dear reviewer

Thank you for accepting our paper. For more improvement, the English in the revised manuscript has been edited by MDPI.

I've attached English-Editing-Certificate.
